

# Multiparameter quantum critical metrology

Giovanni Di Fresco[1*], Bernardo Spagnolo[1,2], Davide Valenti[1] and Angelo Carollo[1]

**1** Department of Physics and Chemistry "Emilio Segrè", Group of Interdisciplinary Theoretical Physics, Università di Palermo, Viale delle Scienze, Ed. 18, I-90128 Palermo, Italy
**2** Radiophysics Department, Lobachevsky State University, Nizhny Novgorod, Russia

⋆ giovanni.difresco01@unipa.it

## Abstract

Single parameter estimation is known to benefit from extreme sensitivity to parameter changes in quantum critical systems. However, the simultaneous estimation of multiple parameters is generally limited due to the incompatibility arising from the quantum nature of the underlying system. A key question is whether quantum criticality may also play a positive role in reducing the incompatibility in the simultaneous estimation of multiple parameters. We argue that this is generally the case and verify this prediction in paradigmatic quantum many-body systems close to first and second order phase transitions. The antiferromagnetic and ferromagnetic 1-D Ising chain with both transverse and longitudinal fields are analysed across different regimes and close to criticality.

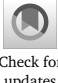

# 1 Introduction

The main purpose of metrology is gaining the best accuracy possible in the estimation of physical parameters, both in classical [1] and quantum systems [2]. Quantum metrology exploits quantum effects to enhance the sensitivity in the estimation, providing advantages in a variety of applications which ranges from gravitational wave detection [3], measuring standards, magnetometry [4], thermometry, imaging [5, 6], navigation, remote sensing, superresolution [7–10] and many more [11]. Most of these applications intrinsically involve the estimation of multiple parameters at the same time, which explains the growing interest in multiparameter quantum metrology [12–14], both theoretically [15–41] and experimentally [42–46].

The extreme sensitivity to small parameter changes is one of the defining characteristic of quantum many-body systems near criticality [47, 48]. The possibility of exploiting this sensitivity in single parameter metrology has attracted a growing interest in the last few years [49–60]. Therefore, a question naturally arises: can the advantages of using interacting many-body systems near criticality be extended to the simultaneous estimation of multiple parameters?

Answering this question is not straightforward, both on a conceptual and computational level. With respect to single parameter quantum metrology, the multiparameter case poses an extra challenge, arising from the very foundation of quantum mechanics: the incompatibility of multiple variables [61, 62]. This results in a trade-off between uncertainties, which complicates in a non-trivial way the quest for the optimal simultaneous measurements already in finite dimensional systems [37, 45, 63]. Extending this problem to a many-body setup in presence of incompatibility is certainly a daunting task.

One way around this task is evaluating the extent to which this incompatibility affects the estimation problem. This can be done efficiently by resorting to a recently introduced quantity called *quantumness* [36, 64]. The quantumness measures the *asymptotic incompatibility* of a multiparameter metrological problem in the limit of an infinite number of copies. The novelty and importance of this approach resides in its simplicity. Indeed, it allows the straightforward evaluation of the estimation's incompatibility from easy-to-compute quantities of the system of interest.

The standard bounds in the accuracy of a quantum multiparameter protocol are given in the form of a matrix inequality for the mean square error matrix by the quantum Cramer-Rao bound (QCRB) [2, 65, 66]. The QCRB is not always tight, due to the aforementioned incompatibility. Instead, the Holevo-Cramer-Rao bound (HCRB) stands out as the ultimate (scalar) bound of multiparameter quantum estimation problems [65], in that it is always achievable in collective measurements on asymptotically large number of copies [67–71]. However, the HCRB, except for few simple cases [72–75], is far from straightforward to compute, even numerically [37].

By contrast, the *quantumness*, denoted by $R_\lambda$[1], is a scalar quantity that can be easily evaluated through the quantum Fisher information matrix (QFIM) $F_Q$ and the mean Uhlmann curvature (MUC) matrix $U$ [76] and quantifies the discrepancy between the HCRB and the QCRB. Its values range in $R_\lambda \in [0, 1]$, with $R_\lambda = 0$ if and only if the two bounds coincide, in which case the multiparameter estimation problem is *asymptotically compatible*. Its maximum value, $R_\lambda = 1$, marks the maximal discrepancy between the QCRB and the HCRB, which in turn signals the maximal incompatibility between the parameters to be estimated, even in the asymptotic limit [77, 78].

In this work, we analyze the compatibility of multiparameter quantum metrology near continuous quantum phase transitions (QPTs) and first order QPTs, using as a main figure of merit the quantumness along with the scaling properties of the QFIM. To this end, we consider

---

[1] The pedix $\lambda$ denotes the set of parameters to be estimated in the metrological protocol

two paradigmatic models: a ferromagnetic and antiferromagnetic Ising chain, both interacting with transverse and longitudinal fields. Moreover, a third model, a spin-1/2 $XY$ chain with transverse field, is also considered in appendix B.

Each multiparameter protocol displays peculiar features related to details of the model, however, when it comes to QPT, the quantumness tipically vanishes as criticality is approached. This can be understood using standard scaling arguments [47]. Close to a continuous phase transition physical quantities are characterised by power-law scalings in the system size $L$, hence one may assume that the quantumness scales as $R_\lambda \sim L^{d_R}$ where $d_R$ is a suitable exponent. However, the upper bound $R_\lambda \leq 1$ [79] is only compatible with *non-positive exponents*, i.e. $d_R \leq 0$. Analogous arguments applied to first order phase transitions lead to similar conclusions, with scalings which are however dependent on the boundary conditions (see scaling analysis section). A further insight is also provided by the definition of the quantumness which reads [79]

$$R_\lambda = \left\| 2iF_Q^{-1} U \right\|_\infty , \tag{1}$$

where $\|X\|_\infty$ denotes the largest eigenvalue of $X$. The inverse dependence of $R_\lambda$ on $F_Q$, which generally diverges at criticality, together with the fact that the MUC may at most diverge with the same rate as $F_Q$ [79], explains the vanishing behaviour of $R_\lambda$. Hence, one may argue that the divergence of $F_Q$, which is the feature that makes critical systems highly attractive for single parameter quantum estimation, is also behind the mechanism responsible for the mitigation of the incompatibility.

## 2 Brief summary of the theoretical background

A system involved in a quantum estimation problem can be described by a family of quantum states $\rho_\lambda$ labelled by a set of parameters $\lambda$, defined in a $p$-dimensional manifold $M$. A multi-parameter quantum estimation problem is a quest for the best accuracy possible in the simultaneous estimation of $\lambda$ [40, 62, 64, 79]. The quantum Cramer-Rao bound (QCRB) provides a lower bound for the mean square errors of the parameters $\lambda$, which can be formally written as [40],

$$\Sigma \geq F_Q^{-1} , \tag{2}$$

where $\Sigma = \text{cov}(\hat{\lambda})$ is the covariance matrix of any locally unbiased estimators $\hat{\lambda}$ of the parameters $\lambda$ and $F_Q$ is the Fisher information matrix, whose components

$$F_{Q_{\mu\nu}} = \frac{1}{2} Tr\left( \rho_\lambda \left\{ L_\mu, L_\nu \right\} \right) , \tag{3}$$

are defined in terms of $\{L_\mu\}_{\mu=1}^p$, a set of self adjoint operators known as *symmetric logarithmic derivatives* (SLD), each satisfying the equation

$$\frac{L_\mu \rho_\lambda + \rho_\lambda L_\mu}{2} = \partial_\mu \rho_\lambda , \tag{4}$$

where $\partial_\mu = \partial / \partial_{\lambda_\mu}$. As mentioned in the introduction, the bound in Eq. (2) is not always tight, unless the following compatibility condition is met [40, 62]

$$U_{\mu\nu} = -\frac{1}{4} \text{Tr}\left\{ \rho_\lambda \left[ L_\mu, L_\nu \right] \right\} = 0 , \quad \forall \mu, \nu , \tag{5}$$

where $U_{\mu\nu}$ is known as *mean Uhlmann curvature* [64,76], a quantity which reduces to the Berry curvature when $\rho_\lambda$ is a family of pure states. The compatibility condition (5) ensures that the

discrepancy between the QCRB and the Holevo-Cramer-Rao bound (HCRB) is zero [40]. The discrepancy between the two bounds can be expressed as

$$D(W) = C_H(W) - \text{tr}\left(W F_Q^{-1}\right), \tag{6}$$

where $W$ is a positive definite weight matrix and $C_H(W)$ is the HCRB [40]

$$\text{tr}\left[W\Sigma\right] \geq \min_{\{X_i\}}\left\{\text{tr}\left[W\Re(V)\right] + \left\|\sqrt{W}\Im(V)\sqrt{W}\right\|_1\right\} = C_H(W), \tag{7}$$

with $\|\cdot\|_1$ being the operator trace norm ($\|\cdot\| = \text{tr}(|\cdot|)$), $V_{i,j} = \text{tr}\left(X_i X_j \rho_\theta\right)$, and the minimization is being performed over the Hermitian matrices $X_i$, which satisfy $\frac{1}{2}\text{tr}\left(\{X_i, L_j\}\rho_\theta\right) = \delta_{i,j}$. This last constraint plays the role of the local unbiasedness condition. It should be pointed out that the minimization performed in Eq. (7) makes really difficult to evaluate the HCRB for systems of interest. The discrepancy in Eq. (6) satisfies [79]

$$0 \leq D(W) \leq \text{tr}\left(W F_Q^{-1}\right) R_\lambda, \tag{8}$$

where $R_\lambda$ is a scalar index, known as *quantumness*, defined as

$$R_\lambda = \left\|2i F_Q^{-1} U\right\|_\infty, \tag{9}$$

with $\|X\|_\infty$ denoting the largest eigenvalue of $X$, and the pedix $\lambda$ specifying the set of parameters to be estimated in the metrological protocol. As already noted, the value of $R_\lambda$ ranges in $[0, 1]$: the limit $R_\lambda = 0$ is equivalent to Eq. (5), and therefore denotes compatibility, whereas $R_\lambda = 1$ marks the maximal incompatibility of the metrological problem. The quantumness obeys a monotonic behaviour with respect to quantum estimation sub-model [77] that could be formalized as follows. If $R_\lambda^{(p)}$ is the quantumness of an estimation model defined by a set of $p$ parameters $\lambda$, and $R_{\tilde{\lambda}}^{(p-1)}$ is the quantumness of the possible sub-model defined by a subset of (possibly reparameterised) $p-1$ parameters $\tilde{\lambda}$, then the following bound holds

$$R_{\tilde{\lambda}}^{(p-1)} \leq R_\lambda^{(p)}. \tag{10}$$

In other words, any multi-parameter estimation protocol is incompatible at least as much as any of its sub-models. This also means that evaluating the quantumness of a full multi-parameter estimation protocol may hide possible compatibilities between some of its parameters. In this sense, it may be more informative to analyse the quantumness of some of its sub-models separately.

In particular, for a two-parameter estimation problem the expression for the quantumness acquires a particularly simple form [36]

$$R_\lambda^{(2)} = \sqrt{\left|\frac{\det(2U)}{\det(F_Q)}\right|}. \tag{11}$$

## 3 Scaling analysis

In this section we provide a scaling analysis of the quantumness close to QCP. This analysis shows that $R_\lambda$ cannot increase close to QCP and generally decreases with a critical exponent which depends both on the property of the critical system and on the chosen parameters. We follow closely the method reported in Ref. [80, 81] that can be applied to study the finite size scaling (FSS) of both continuous and first order quantum phase transition (QPT). We first focus on a continuous QPT.

## 3.1 Continuous phase transition

Let us suppose to have a $d$-dimensional lattice model with linear size $L$, whose Hamiltonian $H(\boldsymbol{\lambda})$ depends on the set of parameters $\{\lambda_\mu\}$. The critical point of a continuous QPT is characterized by scale invariance, and by power-law behaviours of physical quantities which have universal character. Such a universal behaviour emerges between microscopic Hamiltonians which differ by terms, known as irrelevant operators, that become vanishingly small under coarse-graining transformation of the lattice. In such a situation, one can extract information on the universal properties of the system by performing scaling transformations that modify the lattice spacing $a \to \alpha a$. As a consequence, lengths and time rescale as $x \to x\alpha$ and $t \to t\alpha^z$ [47], where $z$ is the dynamical critical exponent. Around the critical point, each local operator can be decomposed in a set of operators, called relevant operators, which dominates the physical property of the system and obey a power-law scaling, $\mathcal{O}_i \to \alpha^{-d_i}\mathcal{O}_i$, where $d_i$ is the operator scaling dimension. If one of the parameter, e.g. $\lambda_\mu$, drives the system close to the criticality, the correlation length of the system is given by $\xi_\mu = (|\lambda_\mu - \lambda_\mu^c|/\lambda_\mu^c)^{-\nu_\mu}$, where $\lambda_\mu^c$ is the critical value of the parameter and $\nu_\mu$ is the correlation length critical exponent.

One can use the scaling behaviours with respect to $\alpha$ in conditions in which the system nearly obeys scale invariance. The most relevant perturbation which breaks the scale invariance dominates the scaling behaviour. For example, if $L \gg \xi_\mu$, the most relevant perturbation is given by $\alpha \sim \xi_\mu^{-1} = (|\lambda_\mu - \lambda_\mu^c|/\lambda_\mu^c)^{\nu_\mu}$. On the other hand, close to criticality in a finite system with size $L \ll \xi_\mu$, the system is dominated by system size effect and $\alpha \sim L^{-1}$ is the most relevant perturbation. The latter is the regime on which we focus.

As for any other physical quantity, one may assume that the quantumness obeys a scaling law $R_\lambda \sim \alpha^{-d_R}$, which in a finite size regime, with $L \ll \xi_\mu$, implies

$$R_\lambda \sim L^{d_R}. \tag{12}$$

As mentioned in the introduction, the upper bound $R_\lambda \leq 1$ [79] is only compatible with a *non-positive exponent*, i.e. $d_R \leq 0$.

Although this argument is quite general, one can make a more detailed analysis on the scaling behaviour of the quantumness, based on the scaling properties of $F_Q$ and $U$ and on the universal properties of the underlying model. To this end, we will assume that the operators $\partial_\mu H$ can be expressed as the sum of local operators i.e. $\partial_\mu H = O_\mu = \sum_x O_\mu(x)$, where $x$ labels a spatial position on the lattice, and that $O_\mu$'s are relevant operators with scaling dimension $d_\mu$'s. For the system in its ground state, $F_{Q\mu\nu}$ and $U_{\mu\nu}$ can be expressed in a compact form [79,82,83]

$$F_{Q\mu\nu} = \frac{2}{\pi} \int_{-\infty}^{+\infty} \frac{d\omega}{\omega^2} \mathcal{S}_{\mu\nu}^+(\omega), \tag{13}$$

$$U_{\mu\nu} = \frac{i}{\pi} \int_{-\infty}^{+\infty} \frac{d\omega}{\omega^2} \mathcal{S}_{\mu\nu}^-(\omega), \tag{14}$$

where $\mathcal{S}_{\mu\nu}^\pm(\omega) := \frac{\mathcal{S}_{\mu\nu}(\omega) \pm \mathcal{S}_{\nu\mu}(\omega)}{2}$ are the symmetric and anti-symmetric parts of the dynamical structure factor $\mathcal{S}_{\mu\nu}(\omega) := \int_{-\infty}^{\infty} dt\, e^{i\omega t} \langle O_\mu(t) O_\nu \rangle$, and $O_\mu(t) := e^{iHt} O_\mu e^{-iHt}$.

The scaling of $F_Q$ close to a critical point can be derived from the symmetric structure factors, which scale as $\int_{-\infty}^{\infty} d\omega\, \mathcal{S}_{\mu\nu}^+ \sim \langle \{O_\mu, O_\nu\} \rangle \sim \alpha^{-d_\mu - d_\nu}$, and $\omega \to \omega\alpha^{-z}$. Thus, from Eqs. (13) we get

$$F_{Q\mu\nu} \to F_{Q\mu\nu} \alpha^{-d_\mu - d_\nu + 2z}. \tag{15}$$

On the other hand, the anti-symmetric structure factor scales as $\int_{-\infty}^{\infty} d\omega\, \mathcal{S}_{\mu\nu}^- \sim \langle [O_\mu, O_\nu] \rangle$, with a dependence on the commutator which scales with an exponent $d_{\mu\nu}^- \leq d_\mu + d_\nu$. Accordingly,

from Eq. (14) we obtain the following scaling for $U_{\mu\nu}$:

$$U_{\mu\nu} \to U_{\mu\nu}\alpha^{-d_{\mu\nu}^- + 2z}. \tag{16}$$

According to Eq. (9) or its two parameter version (Eq. (11)), and in the hypothesis in which the scaling of $F_Q$ and $U$ is dominated by their universal behaviours, we obtain

$$R_\lambda \to R_\lambda \alpha^{-d_R}, \qquad \text{with} \quad d_R = d_{\mu\nu}^- - d_\mu - d_\nu \leq 0. \tag{17}$$

For a system with finite size at the critical point the scale invariance is broken by the system size, which scales as $L \sim \alpha^{-1}$, yielding

$$R_\lambda \sim L^{d_R}, \quad \text{with} \quad d_R < 0. \tag{18}$$

Alternatively, by exploiting the relation (34) (see Appendix A), one can compute the scaling of $F_Q$ from that of the fidelity, as

$$\mathcal{F}(\lambda, \delta\lambda, L) = 1 - \tfrac{1}{8}\sum_{\mu,\nu}\delta\lambda_\mu\delta\lambda_\nu F_{Q\mu\nu} + 0\left(\delta\lambda^3\right), \tag{19}$$

where $\mathcal{F}(\lambda, \delta\lambda, L)$ denotes the fidelity between infinitely close ground states $\psi(\lambda)$ and $\psi(\lambda + \delta\lambda)$ of a system of size $L^d$. By following standard scaling argument [47] any physical observable $\mathcal{O}$ close to a QCP can be expressed in terms of a scaling function $f_\mathcal{O}(\kappa)$ as

$$\mathcal{O} \approx L^{-y_\mathcal{O}} f_\mathcal{O}(\kappa), \tag{20}$$

where $y_\mathcal{O}$ is the scaling dimension of $\mathcal{O}$ and $\kappa = \{\kappa_\mu\}$ is a collection of suitable combinations of parameters $\lambda$ and $L$ as

$$\kappa_\mu = \lambda_\mu L^{y_\mu}. \tag{21}$$

Generalising the arguments in Ref. [80] to multiparameter scenarios we can express the fidelity close to QPT in terms of the rescaled parameterisation as

$$\mathcal{F}(\lambda, \delta\lambda, L) \approx \mathcal{F}(\kappa, \delta\kappa), \tag{22}$$

where the dependence on $L$ is implicit in $\kappa$, and $\delta\kappa$ are variations due to $\delta\lambda$. Now expanding $\mathcal{F}$ in power of $\delta\kappa$ as

$$\mathcal{F}(\kappa, \delta\kappa) = 1 - \sum_{\mu,\nu}\delta\kappa_\mu\delta\kappa_\nu f_{\mu\nu}(\kappa) + o\left(\delta\kappa^3\right), \tag{23}$$

and combining Eq.(19) and Eq.(23), it is possible to obtain the QFIM as

$$F_{Q\mu\nu} \approx 8L^{y_\mu + y_\nu} f_{\mu\nu}(\kappa). \tag{24}$$

Again, the scaling of the quantumness for a two parameter model can be inferred from Eq. (11) and from the scaling of $F_Q$ and $U$. As argued already, the scaling of the determinant of MUC is always bounded above by the scaling of the determinant of QFIM. This can be deduced from the Schrödinger-Robertson uncertainty inequality applied to the SLD [84]

$$\det\left[\frac{1}{2}\text{Tr}\rho\{L_\mu, L_\nu\}\right] \geq \det\left[-\frac{i}{2}\text{Tr}\rho[L_\mu, L_\nu]\right], \tag{25}$$

which, compared to Eqs. (3) and (5), yields

$$\det F_Q \geq \det 2U. \tag{26}$$

If, for simplicity, we assume $F_Q$ in diagonal form, then $\det F_Q \approx L^{2(y_\mu + y_\nu)} f_{\mu\mu} f_{\nu\nu}$, and $\det U \approx L^{2u} f_u$ with $u \leq y_\lambda + y_\sigma$. We find again the scaling of the quantumness as

$$R_{\lambda\sigma} \approx L^{u - (y_\lambda + y_\sigma)} f_R = L^{y_R} f_R, \tag{27}$$

where $y_R \leq 0$.

## 3.2 First order phase transition

It is possible to apply a similar procedure to study first order QPT scaling. The scaling behavior of this kind of QPT is crucially dependent on the boundary conditions [80]. As before, let us assume to have a Hamiltonian $H(\boldsymbol{\lambda})$ dependent of a set of parameters $\boldsymbol{\lambda} = \{\lambda_\mu\}$ and let's study the FSS in proximity of a first order QPT. First order QPT generally arises from level crossing, which only occurs in the infinity-volume limit. For finite size systems, the QPT is characterised instead by avoiding level crossings, whose energy gap rules the FSS behaviour. Following Ref. [81] we define the avoiding level crossing energy gap

$$\Delta_0(L) = \Delta(\boldsymbol{\lambda} = \boldsymbol{\lambda}^c, L), \tag{28}$$

where $\boldsymbol{\lambda}^c$ are critical values of the parameters, and we introduce a set of rescaled parameters which characterize the FSS

$$\tilde{\lambda}_\mu = \frac{E_\mu(\boldsymbol{\lambda}, L)}{\Delta_0}, \tag{29}$$

where $E_\mu(\boldsymbol{\lambda}, L)$ is the energy gap variation due to a change in $\lambda_\mu$ from its critical value $\lambda_\mu^c$. By following a similar argument as in [80] one can derive the following scaling

$$F_{Q_{\mu,\nu}} \sim \frac{(\partial_\mu E_\mu)(\partial_\nu E_\nu)}{\Delta_0^2(L)}. \tag{30}$$

Notice that the divergence of the QFIM with the system size is strongly influenced by the dependence of the energy gap $\Delta_0(L)$. Depending on the type of boundary condition, the gap may vanish exponentially with the system size, i.e. $\Delta_0(L) \sim e^{-aL^d}$ or with a power-law behaviour, $\Delta_0(L) \sim L^{-b}$ [81]. In either case, using the same argument as in the previous section, one can show that the scaling of $U$ is bounded by that of $F_Q$, and the quantumness must scale as

$$R_{\mu\nu} \sim \frac{y_U \Delta_0^2(L)}{(\partial_\mu E_\mu)(\partial_\nu E_\nu)} \le 1. \tag{31}$$

Notice that the dependence of $\Delta_0(L)$ in Eq. (31), is compatible with an either exponential or power-law scaling to zero, depending on the boundary conditions Ref. [81].

# 4 Ising model with transverse and longitudinal fields

We analyse the metrological properties of a 1-D quantum Ising chain with a transverse magnetic field in $x$ and $y$ directions, and a longitudinal magnetic field in $z$ direction. The parameters to be estimated are the coupling constants of the magnetic field, appearing in the Hamiltonian

$$H = -\sum_{i=1}^n \sigma_i^z \sigma_{i+1}^z + h_x \sigma_i^x + h_y \sigma_i^y + h_z \sigma_i^z, \tag{32}$$

where $\sigma_i$ are the Pauli matrices and $n$ is the number of spins. This kind of estimation protocol cannot be interpreted as a canonical interferometric metrological scheme [85]. Rather, this coincides with the standard picture used in single-parameter quantum critical metrology, whereby the Hamiltonian parameters are estimated through the effects they have on corresponding equilibrium state [49, 50, 53, 57]. Therefore, we can find an estimate of the parameters of interest by studying how the properties of the probe states change as the Hamiltonian parameters vary. To analyse in details the compatibility of this model we will consider the quantumness associated to pairs of magnetic field amplitude, which we will denote as $R_{\mu\nu} \equiv R_{\{h_\mu, h_\nu\}}$ with $\mu, \nu \in \{x, y, z\}$.

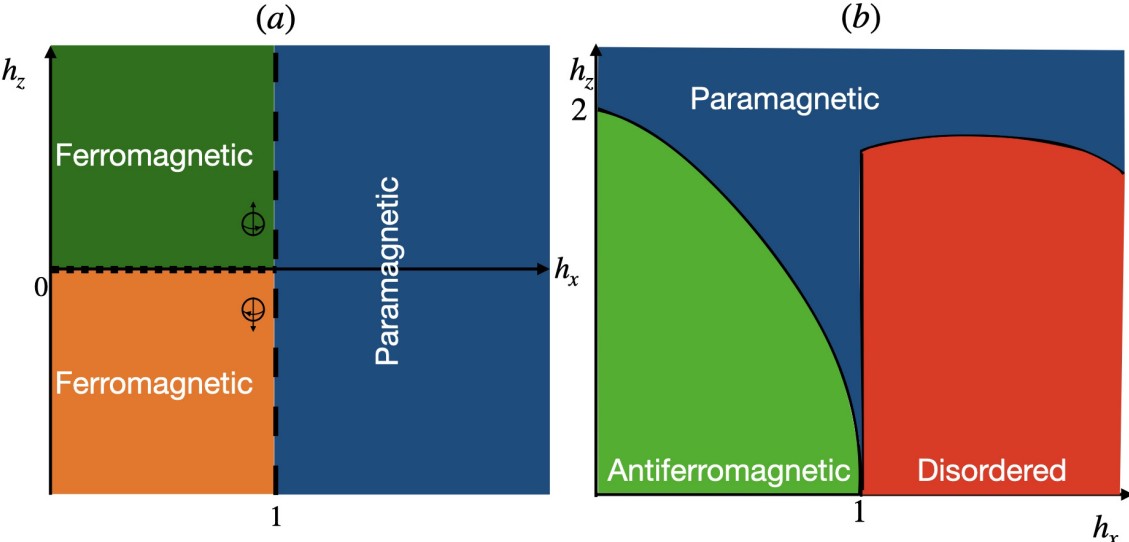

Figure 1: Panel (a): phase diagram of the ferromagnetic 1-D Ising chain with longitudinal and transverse magnetic fields. Panel (b): phase diagram of the antiferromagnetic 1-D Ising chain with longitudinal and transverse magnetic fields.

At $h_z = 0$, $h_y = 0 (h_x = 0)$ and $h_x = 1 (h_y = 1)$ the model undergoes a continuous QPT belonging to the two-dimensional Ising universality class, separating a disordered phase $(h_x > 1)$ from an ordered one $(h_x < 1)$. For any point in $|h_x - h_y| < 1$, the longitudinal field drives a first order QPT along the $h_z = 0$ plane (see panel (a) of Fig. 1). We will limit our study to the zero temperature case, hence we can choose the ground state of Eq. (32) as input probe, which on the one hand allows capturing the features of the QPT, and on the other simplifies the evaluation of QFIM and MUC. Note that the presence of a longitudinal field term in Eq. (32) breaks the integrability of the Hamiltonian and the estimation problem requires a numerical approach. Despite the further complication due to the non-analyticity of the problem, the presence of a longitudinal term $h_z$ allows us to add in the estimation problem a parameter that couples with the order parameter of the second order QPT ($\langle S_z \rangle$). This provides the opportunity to test the role of the order parameter in the estimation problem.

The Hamiltonian in Eq. (32) is numerically diagonalized through the application of the *implicitly restarted Lanczos method*. Due to the lacking of an analytic expression for the ground state of Eq. (32), we will resort to the fidelity approach to calculate the QFIM susceptibility (See appendix A). This approach is a multiparameter generalization of the method used in Ref. [80, 86]. After computing the ground states for two relatively close values of the parameters, the fidelity can be calculated as the overlap between these two states. This procedure is repeated with different pairs of states which are taken progressively away from each other along the $\lambda_i$ direction in the parameter space. Eventually, the fidelity susceptibility is found through a parabolic fitting of the fidelity against $\lambda_i$ [80]. On the other hand, the MUC can be evaluated with a numerical approach similar to that of Ref. [87]. Exploiting the relation with the Berry curvature for pure states, the MUC can be computed through the Bargmann phase [88, 89], which is a version of the Berry phase evaluated on a discretized circuit in the parameter space (see appendix A). The results of these calculations are used to evaluate $R_\lambda$ through its analytic expression (9) across the phase diagram, as displayed in panel (a) of Fig. 2.

A detailed numerical analysis of the quantumness displays an apparent insensitivity across the QPT in $(h_x = 1, h_y = 0, h_z = 0)$. Specifically, we find that for $h_y = h_z = 0$ and $h_x \in (0, 2)$ the quantumness is constantly equal to $R_{xy} = R_{xz} = 0$ and $R_{yz} = 1$.

This trivial behaviour, not shown here, of the quantumness is due to the overwhelming

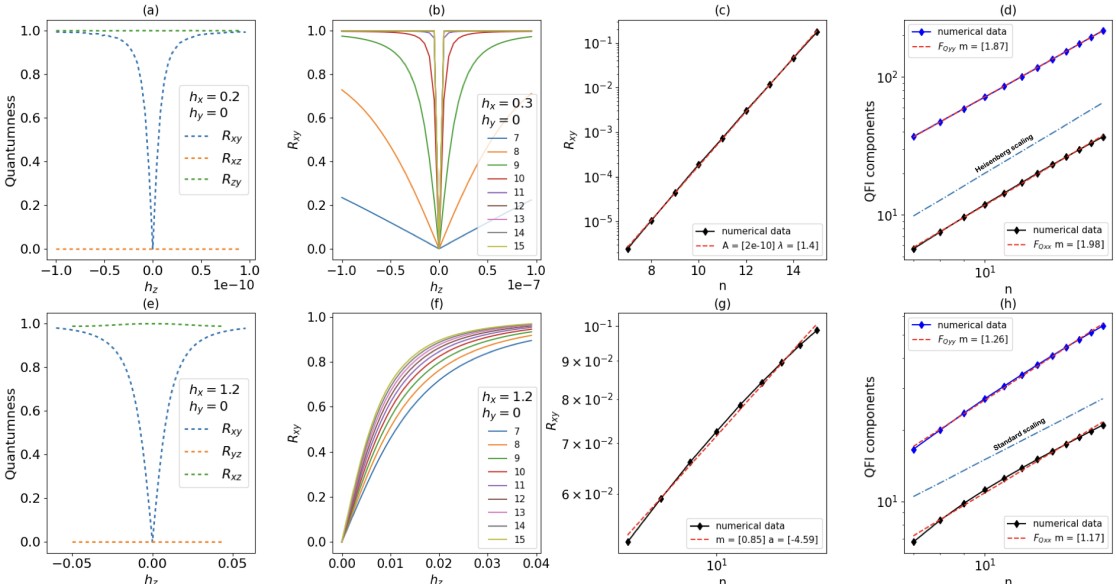

Figure 2: Behaviour of the quantumness and Fisher information in the ordered phase (panels (a-d)) and in the disordered phase (panels (e-h)). In panels (a) and (e) the quantumness is plotted as a function of $h_z$ for $h_x = 0.2$ and $h_x = 1.2$, respectively, with $n = 11$. Panel (b) displays the scaling behaviour of the quantumness for $h_x = 0.3$ as a function of $h_z$. Panel (c) plots in a semi-log scale the quantumness as a function of $n$, very close to the first order QPT ($h_x = 0.3$ and $h_z \sim 10^{-10}$): this shows that $R_{xy}$ decreases abruptly to zero in $h_z = 0$, with a rate which scales exponentially with $n$. Analogously, panel (f) displays the dependence of $R_{xy}$ on $h_z$ and panel (g) its dependence on $n$ close to $h_z = 0$ and for $h_x = 1.2$ (log-log scale): the quantumness decreases to zero, but in absence of a first order QPT it vanishes with a rate which scales as a power-law in $n$. In the log-log plots in panel (d) and (h) the scaling behaviour of the components of the QFI is shown with respect to the number of spins n. In panel (d) it is shown that for $h_x = 0.95$ both the components are very close to a Heisenberg scaling. Otherwise, in panel (h) in absence of QPT ($h_x = 1.2$) both the components are far from a Heisenberg scaling and closer to a normal one. Notice that the fitting parameter $m$ is the slope of the linear fitting, while $A$ and $\lambda$ are the amplitude and the coefficient of the exponential fitting, respectively. In panel (d) and (h), the blue dashed lines are guides for the eye corresponding to Heisenberg and standard scalings, respectively.

effects of the first order phase transition, which hides the dependence of $R_{\mu\nu}$ on the continuous QPT.

On the other hand, panels (a) and (e) of Fig. 2 display the behaviour of $(R_{xy}, R_{xz}, R_{yz})$ versus the longitudinal field $h_z$, for $h_y = 0$ and $h_x$ fixed. Also in this case $R_{xz}$ and $R_{yz}$ are insensitive to the field, both in the ordered phase (panel (a), $h_x = 0.2$) and in the disordered one (panel (e), $h_x = 1.2$). Panel (a) shows that the only component sensitive to the first order QPT is $R_{xy}$, with a sharp reduction to zero across $h_z = 0$, for $h_x < 1$ (ordered phase). However, panel (e) of Fig. 2 shows that $R_{xy}$ goes to zero also for $h_x > 1$ (region in which no first order phase transition is present). Despite the apparent similarity in the behaviours of $R_{xy}$ for $h_x > 1$ and $h_x < 1$, their behaviour is qualitatively different in the two regions, due to the presence (in the ordered phase) and absence (in the disordered phase) of a first order QPT. In order

to show that $R_{xy}$ is actually sensitive to the first order QPT, we show in panels (b) and (f) of Fig. 2 the scaling behaviour with the system size of $R_{xy}$ in the two different regions.

In panel (b), when the system crosses $h_z = 0$ with $h_x < 1$, the first order QPT occurs and $R_{xy}$ goes abruptly to zero with a rate which grows exponentially with the number of spins, as shown explicitly in panel (c). On the contrary, panel (f) displays the behaviour of $R_{xy}$ across $h_z = 0$ in the disordered region ($h_x > 1$), where no first order QPT occurs: here the quantumness goes smoothly to zero with a rate which is power-law dependent on the system size (see also panel (g) for a power-law fitting).

From a metrological point of view it is also meaningful to study closely the behaviour of the QFI and its scaling near the phase transition. Panels (d) and (h) in Fig. 2 show that the QFI has different scaling behaviours in the two regions of the phase diagram. In panel (d) it is shown that in the ferromagnetic region, with $h_x$ near 1, both the $x$ and $y$ components of the QFI have a scaling very close to the Heisenberg limit, which allows to perform precise estimation in each direction. Otherwise, in the paramagnetic region (panel (h)) both the components of the QFI have a less enhanced scaling, closer to the standard quantum limit.

The behaviour of the two components of the inverse QFI is not far from that of the reciprocal QFI since, despite their presence, the off-diagonal elements in the QFI matrix are order of magnitude smaller than the diagonal elements.

It is worth mentioning that analyzing the quantumness associated with all the components of the magnetic field does not provide useful information for this system. Panel (a) and panel (e) of figure 2 show that the quantumness associated with at least one pair of components of magnetic fields is always maximal. In fact, from Eq. (10) it is straightforward to deduce that the quantumness of the complete set of parameters is always maximal, i.e. $R_{xyz} = 1$.

## 5   Antiferromagnetic Ising chain

Due to the presence of a longitudinal magnetic field, the antiferromagnetic Ising chain has different properties from those of the ferromagnetic one. In fact, the Hamiltonian

$$H = \sum_i \sigma_i^z \sigma_{i+1}^z - h_x \sigma_i^x - h_y \sigma_i^y - h_z \sigma_i^z \tag{33}$$

is characterized by a completely different phase diagram, as we can see from panel (b) of Fig. 1 [90]. The main difference, in the region of interest, lies in a stable antiferromagnetic phase for values of the longitudinal magnetic field different from zero with a consequent line of continuous QPTs in which $h_z \neq 0$. This shifts our focus to a region of the parameter space in which more than one component of the magnetic field is non-vanishing.

To map the Hamiltonian in Eq. (33) into a ferromagnetic model we need a staggered magnetic field [91], which justifies the differences between the models. We notice that the phase diagram in panel (b) of Fig. 1 can be derived through the fidelity approach [90]. In Ref. [90] it is shown that at the phase transition from an antiferromagnetic to a paramagnetic order, the $x$ component of the QFI matrix exhibits a maximum. Here we study closely the scaling behaviour of the QFI at the transition point and how it affects the compatibility index. We also use only even numbers of spins, to avoid frustration due to the antiferromagnetic nature of the chain, and periodic boundary conditions. As for the ferromagnetic scenario, $R_{xy}$ is, again, the only component of the quantumness sensitive to the phase transition. However, the different properties of the model affects profoundly the behaviour of the quantumness, leading to a completely different dependence on the parameters. Fig. 3 shows that $R_{xy} < 1$ in all the range of the parameters evaluated and that at the phase transition the critical behaviour of the QFI makes the quantumness vanish as the size of the chain increases. In panel (a) of Fig. 4 it is

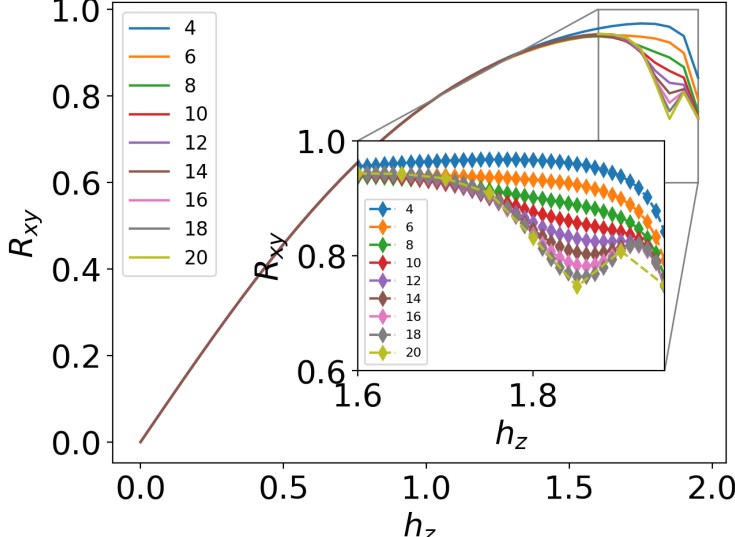

Figure 3: The compatibility index $R_{xy}$ for different values of the chain size and $h_x = 0.2$ as a function of the longitudinal field $h_z$. The inset shows the behaviour of the compatibility index near the critical point.

shown the scaling behaviour of the QFI for $h_x = 0.5$. The $x$ component exhibits a maximum and it reaches a Heisenberg scaling (QFI$\sim N^2$), whereas the $y$ component at the critical point has a standard quantum scaling (QFI$\sim N$). So, at criticality, the system reaches the highest precision in one of the two components while displaying asymptotic compatibility with the other component. Panel (b) of Fig. 4 shows the dependence on the system size of the determinants of QFI and MUC, which both display a power-law scaling. Since the QFI has a scaling higher than the MUC, from Eq. (9) we can extrapolate that the quantumness asymptotically vanishes at the criticality.

# 6 Conclusion

In this work we have analyzed the performance of multiparameter quantum critical estimation protocols focusing on the role of QPT in mitiganting the incompatibility among parameters. From the prototypical models analyzed, in both first and second order QPT, a common feature emerging is the strong dependence of $R_\lambda$ on criticality, and a general influence of QPT in reducing the incompatibility. In a two-parameter magnetometry model with a 1D Ising chain, the sensitivity of the quantumness to a first order QPT is numerically demonstrated. Indeed, the exponential scaling of $R_\lambda$ represents a signature of the first order QPT. A similar setup, in an antiferromagnetic scenario, displays an asymptotic compatibility at the critical point, demonstrated by a vanishing behaviour of $R_\lambda$.

Our work strongly suggests that quantum critical metrology provides a promising framework for multi-parameter estimation. One of the desirable features of critical metrology, i.e. the divergence of the Fisher information, comes with an extra advantage in the multi-parameter scenario: criticality may help in mitigating the incompatibility. The latter is one of the main drawback in quantum multi-parameter metrology, which makes the estimation challenging both on a computational and a conceptual level. Our approach opens up the possibility to explore multi-parameter metrology in many-body setups using easy-to-compute figures of merit, thereby paving the way to fundamental theoretical advances and technological applications.

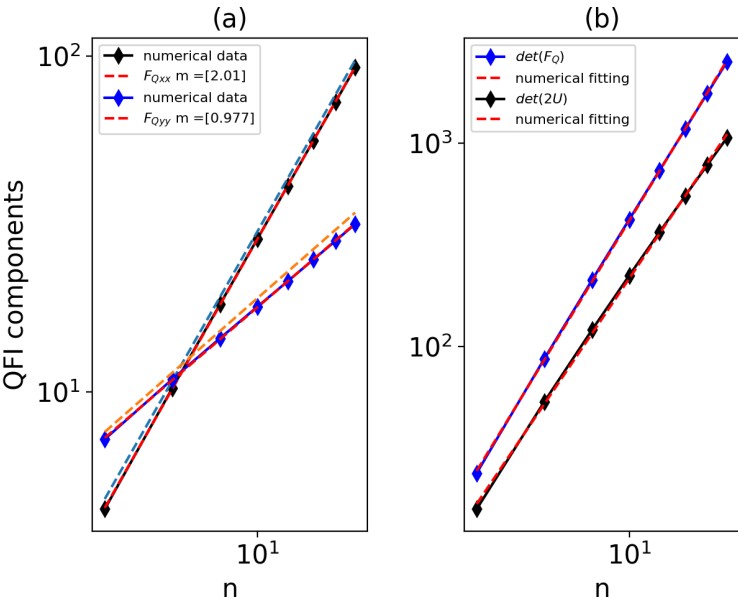

Figure 4: In panel (a) the log-log plot of the diagonal components of the quantum Fisher information whit $h_x = 0.5$ at the critical point. In blue the scaling of the $y$ component, with a slope $m = 0.98$, and in black that of the $x$ component, with a slope $m = 2.01$, as a function of the number of spins $n$. In the log-log plot in panel (b) it is shown in blue the scaling behaviour of the quantum Fisher information matrix determinant at the critical point for $h_x = 0.5$ as a function of the number of spins $n$, with a slope $m = 3.09$; in black the scaling behaviour of twice the mean Uhlmann curvature determinant at the critical point for $h_x = 0.5$, with a slope $m = 2.784$. In panel (a), the blue and orange dashed lines are guides for the eye corresponding to Heisenberg and standard scalings, respectively.

# Acknowledgements

GDF, BS, DV, AC acknowledge the support of the Italian Ministry of University and Research (MUR). BS acknowledges also the support of the Government of the Russian Federation through Agreement No. 074-02-2018-330 (2)

**Author contributions**   **GDF:** Conceptualization; Data curation; Formal analysis; Investigation; Methodology; Software; Validation; Visualization; Roles/Writing - original draft; Writing - review & editing.
**BS**: Conceptualization; Formal analysis; Funding acquisition; Investigation; Methodology; Writing - review & editing.
**DV**: Conceptualization; Formal analysis; Funding acquisition; Investigation; Methodology; Writing - review & editing.
**AC**: Conceptualization; Data curation; Formal analysis; Funding acquisition; Investigation; Methodology; Software; Validation; Roles/Writing - original draft; Writing - review & editing.

# A   Numerical procedure

In the case of non-integrable system, the lacking of closed form expressions for the ground states prevents the calculation of the QFIM and MUC through the SLD. Instead, one can evalu-

ate the QFIM through the fidelity susceptibility, by exploiting the following relation [66,79,86]

$$F_{Q\mu\nu} = -4\partial_\mu\partial_\nu \mathcal{F}[\rho_\lambda, \rho_{\lambda+\delta}], \tag{34}$$

where $\delta$ is a small variation of the parameters along the directions $\lambda_\mu$ and $\lambda_\nu$, and $\mathcal{F}[\rho,\sigma] = \mathrm{Tr}\left(\sqrt{\sqrt{\rho}\,\sigma\sqrt{\rho}}\right)$ is the quantum Uhlmann fidelity [92] beetween $\rho$ and $\sigma$, which for pure states reduces to the state overlap $\mathcal{F}[\psi,\psi'] = |\langle\psi|\psi'\rangle|$.

Similarly, when only pure states are involved, the MUC coincides with the Berry curvature [79], i.e.

$$U_{\mu\nu} = i\langle\partial_\mu\psi|\partial_\nu\psi\rangle - i\langle\partial_\nu\psi|\partial_\mu\psi\rangle. \tag{35}$$

In turns the Berry curvature can be thought of as the geometric phase per unit area on an infinitesimal loop in the parameter space [79], and it can be evaluated with numerical methods specifically designed for geometric phases [87]. This numerical methods consists in the evaluation of a discretised version of the Berry phase, namely the Bargmann phase [87–89], which is defined as

$$\Phi = \arg\{\prod_{i=0}^{N-1} \langle\psi_i|\psi_{i+1}\rangle\}, \tag{36}$$

where $\{\psi_i\}_{i=0}^{N-1}$ (with $\psi_N = \psi_0$) is a set of states lying on the vertices of a discrete close loop in the parameter space. The calculation of the MUC in a given point $\lambda$ of the parameter space is obtained via the Bargmann phase per unit area evaluated on an infinitesimal loop, i.e.

$$U_{\mu\nu}(\lambda) = \lim_{\delta A\to 0} \frac{\Phi_{\mu\nu}(\lambda)}{\delta A}, \tag{37}$$

where $U_{\mu\nu}(\lambda)$ is the matrix element of the MUC, $\Phi_{\mu\nu}(\lambda)$ is the Bargmann phase calculated in an infinitesimal loop centred on $\lambda$ and lying on the plane identified by the parameters $\lambda_\mu$ and $\lambda_\nu$, and $\delta A$ is the area of the loop. An example is shown in Fig. 5, where the states picked for the computation are on the vertices of the infinitesimal rectangle of sides $\delta\lambda_\mu$ and $\delta\lambda_\nu$. Moreover, to improve the numerical stability of the value of $U_{\mu\nu}$, we exploit the linear dependence of $\Phi_{\mu\nu}(\lambda)$ on $\delta A$ and evaluate $U_{\mu\nu}$ through a linear fitting of $\Phi_{\mu\nu}(\lambda)$ against $\delta A$.

## B  Ground state rotation of the XY spin chain

We report here on a model that, unlike the two discussed in sections 4 and 5, can be interpreted as a canonical interferometric model, in which the parameters to be estimated are introduced through a unitary operator. We show that even in this scenario the quantmuness is strongly affected by criticality. The model analyzed is a XY spin chain, whose Hamiltonian is

$$H = -\sum_{i=-M}^{M} \left(\frac{(1+\gamma)}{2}\sigma_i^x\sigma_{i+1}^x + \frac{(1-\gamma)}{2}\sigma_i^y\sigma_{i+1}^y + \lambda\sigma_i^z\right), \tag{38}$$

where the sigmas are the Pauli matrices, $\gamma$ is the anisotropic parameter, and $\lambda$ is the strength of the external field. We limit our analysis to a region of the phase diagram with $\gamma \in (0,1]$, in which the criticality is at $\lambda_c = 1$ and belongs to the Ising universality class [48,93]. Moreover,

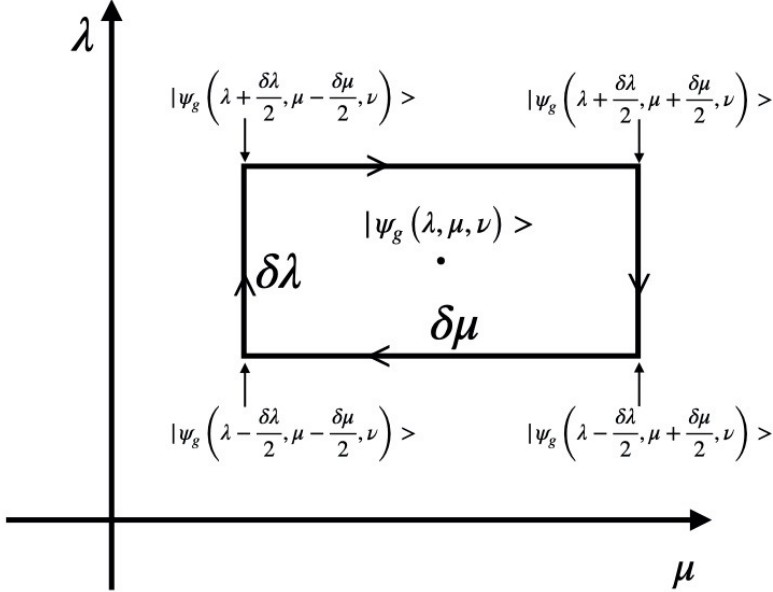

Figure 5: Rectangular circuit in the parameter space.

we only consider the case $T = 0$, in order to capture the essential behaviour of quantum phase transitions. Finally, we assume that the input probe of the estimation is

$$\rho_0 = |\psi_g\rangle\langle\psi_g|, \tag{39}$$

where $|\psi_g\rangle$ is the ground state of Eq. (38). The set of parameters to be estimated is $\boldsymbol{\varphi} = \{\varphi_x, \varphi_y, \ \varphi_z\}$ with

$$\rho_\varphi = U_\varphi^\dagger \rho_0 U_\varphi \quad \text{and} \quad U_\varphi = e^{i\left(\varphi_x S_x + \varphi_y S_y + \varphi_z S_z\right)}, \tag{40}$$

where $\varphi_\mu$ with $\mu = \{x, y, x\}$ are the angles by which the probe state is rotated and $S_\mu = \sum_i \sigma_i^\mu/2$ are the corresponding global spin operators. This unitary transformation can be thought of as the result of adiabatic variation of the parameter $\varphi$ in the system Hamiltonian $U_\varphi^\dagger H U_\varphi$. This coincides with the standard picture used in single-parameter quantum critical metrology, whereby the Hamiltonian parameters are estimated through the effects they have on the corresponding equilibrium state [49, 50, 53, 57]. Alternatively, this unitary transformation can be the result of a dynamical evolution applied to the initial probe state $\rho_0$. In this sense the protocol bears close similarity with the standard interferometric paradigm of quantum metrology [85]. We exploit the unitary symmetry of the problem, thus confining ourselves to the estimation around the point where $\varphi_x = \varphi_y = \varphi_z = 0$. For a pure state probe the SLD [62, 94] is easily calculated, yielding in our case

$$L_\mu = 2\partial_{\varphi_\mu}\rho = 2i\left[S_\mu, \rho\right], \quad \mu = \{x, y, z\}, \tag{41}$$

which in turn leads to the following expressions for the matrix elements of QFIM and MUC

$$F_{\mu\nu} = 4C\left(S_\mu, S_\nu\right), \tag{42}$$

$$U_{\mu,\nu} = -i\text{Tr}\left(\rho\left[S_\mu, S_\nu\right]\right), \tag{43}$$

where $C\left(S_\mu, S_\nu\right)$ is the covariance between the two spin operators.
By exploiting the analytical expressions for the correlation and the expectation values of the $S_\mu$'s for the XY model (see Refs. [93, 95, 96]), the quantumness in Eq. (9) is readily evaluated.

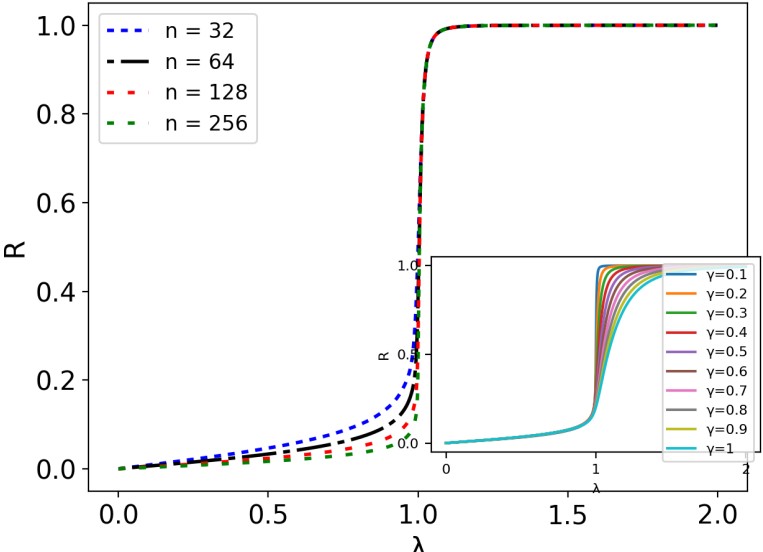

Figure 6: The compatibility index $R(n, \lambda)$ for the $XY$ Ising chain at different values of $n$, up to $n = 256$, with $\gamma = 0.2$ as a function of $\lambda$. Inset: the compatibility index $R(\gamma, \lambda)$ for the $XY$ Ising chain, for different values of $\gamma \in (0, 1]$ and $n = 64$, as a function of $\lambda$. The behaviour of R in the parametric regions appears to sharpen as $\gamma$ gets closer to 0. This effect can be seen in the inset of Fig. 6, that displays the different behaviour of R as $\gamma$ varies in $(0, 1]$, with $n$ fixed.

In Figure 6 the behaviour of the quantumness $R_\varphi$ is shown for different sets of the Hamiltonian parameters and for different numbers of spins. In all the configurations the values of $R_\varphi$, close to zero in the ferromagnetic region, increase gradually for $\lambda < \lambda_C (= 1)$. When the critical point $\lambda = \lambda_C$ is reached the quantumness abruptly saturate to its maximum value $R_\varphi = 1$. Hence, the system goes from maximal incompatibility in the paramagnetic phase to a relatively compatible situation in the ferromagnetic phase, with a more pronounced transition as the number of spins increases. Intuitively, the high compatibility in the ferromagnetic region can be ascribed to the multipartite entanglement of $\rho_0$. Indeed, the form of $\rho_0$ in the ferromagnetic phase bears close similarity to the density matrix of the GHZ states [48], which are optimal probes for multi-parameter quantum magnetometry [3].

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
