# Peer review of "Multiparameter quantum critical metrology"

_SciPost Physics, doi:SciPost Phys. 13, 077 (2022)_

## Round 1 · Referee Report · Anonymous (Referee 1) · 2022-4-12

Strengths

1- The paper is clearly written. 2- The literature is properly cited. 3- The numerics is of good quality, and well presented.

Weaknesses

The results are incremental with respect to the existing literature.

Report

In their paper, the authors study the estimation of non-commuting Hamiltonian parameters in the ground state of one-dimensional spin models, in particular in the vicinity of phase transitions. They show, from general scaling arguments, that the incompatibility -- as quantified by the so-called quantumness -- vanishes at the critical point. They verify this general result by explicit numerical computations on specific spin models. Overall, the paper is well written, and contains appropriate reference to the literature. The main weakness is that their main result is a straightforward consequence of basic scaling arguments, and therefore, to my opinion, does not represent the kind of breakthrough expected for papers published in Scipost Physics. However, provided the following points are clarified, it could be published in Scipost Physics Core.

Specific question: - It appears on Fig. 2 that the quantumness R diverges with n, while according to the authors it should go to 0. Could the authors clarify this point? - There seem to be a typo in Eq.(11): F should be F_Q.

  • validity: good
  • significance: ok
  • originality: ok
  • clarity: high
  • formatting: good
  • grammar: excellent

Author:  Giovanni Di Fresco  on 2022-05-09  [id 2449]

(in reply to Report 1 on 2022-04-12)
Category:
answer to question
reply to objection

We would like to thank the referee for reviewing the paper and for his/her constructive comments. The referee's main criticism regards the straightforwardness of the main result which is obtained through simple scaling arguments applied to the quantumness. While we agree with the referee on the simplicity of the derivation, we would like to point out that in itself simplicity is certainly an added value rather than a weakness, when simplicity is not triviality.

Indeed, the central result of the manuscript is far from trivial and are likely to have a strong impact, for their implications in both quantum metrology and in quantum statistical mechanics. Let us stress that multi-parameter quantum metrology has seen an impressive surge of interest in last few years, testified by a plethora of experimental and theoretical results recently published in high impact journals. Moreover the use of critical phenomena in (single parameter) quantum metrology is now a widely accepted paradigm which may provide enhanced sensitivity in parameter estimation.

To the best of our knowledge, this manuscript is the very first work which addresses the role of quantum criticality in multi-parameter estimation theory on a general footing, and it does it by combining in an original way two key ingredients: the quantumness and the scaling hypothesis. The general scaling behaviour of the quantumness has never been analysed in previous works, and in particular the general mitigation of the compatibility, as a consequence of this scaling, has never been pointed out before. The fact that this behaviour can be deduced from simple scaling argument emphasises even more the relevance of this result, unveiling at once the wide applicability of the ideas explored in the manuscript.

Let us stress once more the conceptual and practical implications of these ideas. On the conceptual side, we show that critical systems may often display an in-built mechanism which limits the effect of incompatibility. This mechanism relies on the very same feature that makes critical systems highly attractive for single parameter quantum estimation: the high rate of divergence of the Fisher Information. We show that close to criticality, while the accuracy in parameter estimation grows as the system size grows, the incompatibility asymptotically vanishes. In this way we not only demonstrate with a scaling argument the reason why we should expect a vanishing quantumness close to criticality, but also provide an intuitive explanation as to why this happens. Indeed, we connect the divergence of the Fisher Information as the main intuitive reason which explains the reduction to zero of the incompatibility.

On the practical side, we manage to demonstrate the asymptotic behavior of the Holevo Cramer Rao bound, bypassing at once the need to perform daunting optimization procedures usually required in standard multi-parameter accuracy bounds, opening up the possibility of applying this procedure to a wide variety of many body setups.

We hope that in the light of the above argument the referee would like to reconsider his/her opinion on the manuscript and warrant publication in SciPost Physics.

Regarding the two specific questions.

Question 1:"- It appears on Fig. 2 that the quantumness R diverges with n, while according to the authors it should go to 0. Could the authors clarify this point?"

Answer: The referee's comment most likely refers to panels (c) and (g) of Fig. 2, where we report the scaling behaviour of the quantumness with respect to the number of spins in the chain. These two panels highlight the different rate with which the quantumness goes to zero, approaching $h_z = 0$, in two regions: (c) in the ferromagnetic region and (g) in the paramagnetic one. For example, the plot shown in (c) is performed for values of the parameter $h_z$ in the order of $10^{-9}-10^{-10}$ which is close, but not exactly equal, to the critical value $h_z=0$. For these values of the parameters and for the number of spins considered one can appreciate the exponential dependence (linear in the semi-log scale). With a higher number of spins this dependence will change, saturating to the value $R=1$, as can be deduced from panel (b). Essentially, in this panel we focus on the rate of convergence of $R$ from $1$ to $0$ in the neighborhood of $h_z=0$ as a function of number of spins $N$. Similarly, panel (g) shows the algebraic dependence of this rate as a function of N in the paramagnetic region.

Question 2: "- There seem to be a typo in Eq.(11): F should be $F_Q$."

Answer: We thank the referee for spotting this typo and we will correct Eq. (11) in the final version of the paper.

---

## Round 1 · Referee Report · Anonymous (Referee 2) · 2022-4-24

Report

In this manuscript the authors discuss multiparameter quantum estimation in critical systems. Critical systems have been already shown to be a resource for quantum metrology, as the QFI corresponding to the estimation of a parameter driving the phase-transition, typically diverges near criticality. In this work the authors exploit a recent measure of quantumness/compatibility to discuss the multiparameter properties near to a phase transition. They first provide a very nice and general explanation showing how this measure of incompatibility vanishes at the critical point; then they discuss in more detail two paradigmatic examples (antiferromagnetic and ferromagnetic 1-D Ising chain with both transverse and longitudinal fields). The results are definitely original and relevant for the research communities working on quantum metrology and quantum many-body systems. I am thus very inclined in recommending it for publication in SciPost. However before being published I think that the manuscript could be improved if the authors will address some (minor) remarks that I will detail in the following

  • I understand that the authors evaluate the quantumness parameter R only for couple of parameters exploiting the property in Eq. (10). However I think it would be interesting to have the information also on the overall (three-parameter) quantumness. Is it always equal to the maximum of the two-parameter R that has benn already evaluated?

  • I have found some of the figures (for example panels (d) and (h) of Fig. 2, or Fig. 4) not completely clear, or at least not self-explanatory. I think it would be more useful in the legend to make clear to what QFI matrix elements the points and the curves correspond to, instead of writing explicitly the results of the numerical fits (in particular I would suggest to put not much emphasis on the values of the intercepts values a of these linear fits).

  • Regarding to these fits, I have found some claims on the corresponding results not very clear/accurate. For example the authors write that in panel (d) one observes Heisenberg scaling, while in panel (h) a “normal scaling”. I think that these conclusions are drawn as the angular coefficients of the numerical linear regressions in the two panels are somehow close respectively to m=2 and m=1. However one does not obtain exactly these results and thus one should be more careful with these claims. For example also in panel (h) one obtains m=1.26 which is, I would say, definitely larger than m=1. In general I would suggest to the authors to — rephrase the corresponding sentences in the caption and in the main text — to add in these plots some lines with slopes corresponding to standard quantum limit scaling and Heisenberg scaling as a guide to the eye, in order to help the reader in observing the transition from one regime to the other one.

  • validity: -
  • significance: -
  • originality: -
  • clarity: -
  • formatting: -
  • grammar: -

Anonymous on 2022-05-09  [id 2450]

(in reply to Report 2 on 2022-04-24)
Category:
answer to question

We would like to thank the referee for reviewing the manuscript and for his/her appreciation of our work. The referee is extremely positive on the originality and relevance of the work for the research community. He/she is "very inclined in recommending it for publication in SciPost".

In the following we will address point by point the remarks of the referee.

Remark 1:

I understand that the authors evaluate the quantumness parameter $R$ only for couple of parameters exploiting the property in Eq. (10). However I think it would be interesting to have the information also on the overall (three-parameter) quantumness. Is it always equal to the maximum of the two-parameter $R$ that has been already evaluated?

Answer: We thank her/him for the interesting question. As can be deduced from the inequality (10), a 3-parameter quantumness is always lower-bounded by any 2-parameter quantumness obtained with a subset of the 3 parameters. In all the models examined we always observed a two-parameter quantumness which remains saturated at $R=1$ across the region of criticality (see $R_{zy}$ Fig.2 (a) and $R_{xz}$ in panel (e) for example). Since, $R=1$ is the maximum value of the quantumness, this immediately implies that the 3-parameter quantumness remains saturated to $R=1$ in all the models examined. Hence, on account of this trivial behaviour we have not included it in the work as it does not convey any relevant information. We will clarify this aspect in the next version of the manuscript.

Remark 2:

I have found some of the figures (for example panels (d) and (h) of Fig. 2, or Fig. 4) not completely clear, or at least not self-explanatory. I think it would be more useful in the legend to make clear to what QFI matrix elements the points and the curves correspond to, instead of writing explicitly the results of the numerical fits (in particular I would suggest to put not much emphasis on the values of the intercepts values a of these linear fits).

Answer: We thank the referee her/him for the comments on the figures that helped us to present our work much more clearly. We will amend the legend to make the figures more self-explanatory, in accordance to the referee suggestion. We have edited the coming version of the manuscript by adding a more clear reference to the QFI component.

Remark 3:

Regarding to these fits, I have found some claims on the corresponding results not very clear/accurate. For example the authors write that in panel (d) one observes Heisenberg scaling, while in panel (h) a “normal scaling”. I think that these conclusions are drawn as the angular coefficients of the numerical linear regressions in the two panels are somehow close respectively to m=2 and m=1. However one does not obtain exactly these results and thus one should be more careful with these claims. For example also in panel (h) one obtains m=1.26 which is, I would say, definitely larger than m=1. In general I would suggest to the authors to — rephrase the corresponding sentences in the caption and in the main text — to add in these plots some lines with slopes corresponding to standard quantum limit scaling and Heisenberg scaling as a guide to the eye, in order to help the reader in observing the transition from one regime to the other one.

Answer: We are again grateful to the referee for his/her remark. In the coming version of the manuscript we are changing the text and the caption to a more accurate description of the scaling rate. We agree that the claim as it currently stands may lead to confusion the reader. Following the suggestion of the referee we are adding lines as a guide for the eye, with the slopes corresponding to normal and Heisenberg scalings.

---

## Round 2 · Referee Report · Anonymous (Referee 3) · 2022-6-1

Report

The authors have satisfactorily replied to all the comments and I am happy to recommend the manuscript for publication in SciPost.

Requested changes

I have only a minor suggestion: in one of the added paragraphs at the end of Sec. 4, in Fig. 2 panel (h) and in its caption, the authors refer to "normal quantum limit" and to "normal scaling" . I would suggest to replace them with the terms "standard quantum limit" and "standard quantum scaling".

---

## Round 2 · Referee Report · Anonymous (Referee 4) · 2022-7-7

Report

The authors have provided satisfactory answers to the Referees' comments. I recommend publication of the manuscript in its present form. I am now convinced that the results presented are relevant enough to be published in SciPost Physics.

---

## Round 2 · Author Response

Dear Editor, We would like to thank you for handling the manuscript and the referees for their insightful comments that have helped us to improve the quality of our work. We are grateful to the first referee for his/her constructive comments and his positive feedback on the quality of presentation. We would like to thank the second referee for his/her appreciation of our work. He/she is extremely positive on the originality and relevance of the work for the research community and he/she is thus “ very inclined in recommending it for publication in SciPost.”

We have revised the manuscript to address each of the referees comment to improve the quality of presentation.

A detailed reply to these comments is given below, illustrating the relative changes made in the manuscript. Yours Sincerely Giovanni Di Fresco, on behalf of all authors.

Reply to Referee 1:

We would like to thank the referee for reviewing the paper and for his/her constructive comments. The referee's main criticism regards the straightforwardness of the main result which is obtained through simple scaling arguments applied to the quantumness. While we agree with the referee on the simplicity of the derivation, we would like to point out that in itself simplicity is certainly an added value rather than a weakness, when simplicity is not triviality.

Indeed, the central result of the manuscript is far from trivial and are likely to have a strong impact, for their implications in both quantum metrology and in quantum statistical mechanics. Let us stress that multi-parameter quantum metrology has seen an impressive surge of interest in last few years, testified by a plethora of experimental and theoretical results recently published in high impact journals. Moreover the use of critical phenomena in (single parameter) quantum metrology is now a widely accepted paradigm which may provide enhanced sensitivity in parameter estimation.

To the best of our knowledge, this manuscript is the very first work which addresses the role of quantum criticality in multi-parameter estimation theory on a general footing, and it does it by combining in an original way two key ingredients: the quantumness and the scaling hypothesis. The general scaling behaviour of the quantumness has never been analysed in previous works, and in particular the general mitigation of the compatibility, as a consequence of this scaling, has never been pointed out before. The fact that this behaviour can be deduced from simple scaling argument emphasises even more the relevance of this result, unveiling at once the wide applicability of the ideas explored in the manuscript.

Let us stress once more the conceptual and practical implications of these ideas. On the conceptual side, we show that critical systems may often display an in-built mechanism which limits the effect of incompatibility. This mechanism relies on the very same feature that makes critical systems highly attractive for single parameter quantum estimation: the high rate of divergence of the Fisher Information. We show that close to criticality, while the accuracy in parameter estimation grows as the system size grows, the incompatibility asymptotically vanishes. In this way we not only demonstrate with a scaling argument the reason why we should expect a vanishing quantumness close to criticality, but also provide an intuitive explanation as to why this happens. Indeed, we connect the divergence of the Fisher Information as the main intuitive reason which explains the reduction to zero of the incompatibility.

On the practical side, we manage to demonstrate the asymptotic behavior of the Holevo Cramer Rao bound, bypassing at once the need to perform daunting optimization procedures usually required in standard multi-parameter accuracy bounds, opening up the possibility of applying this procedure to a wide variety of many body setups.

We hope that in the light of the above argument the referee would like to reconsider his/her opinion on the manuscript and warrant publication in SciPost Physics.

Regarding the two specific questions.

Question 1:"- It appears on Fig. 2 that the quantumness R diverges with n, while according to the authors it should go to 0. Could the authors clarify this point?"

Answer: The referee's comment most likely refers to panels (c) and (g) of Fig. 2, where we report the scaling behaviour of the quantumness with respect to the number of spins in the chain. These two panels highlight the different rate with which the quantumness goes to zero, approaching h_z = 0, in two regions: (c) in the ferromagnetic region and (g) in the paramagnetic one. For example, the plot shown in (c) is performed for values of the parameter h_z in the order of 10^-9-10^-10 which is close, but not exactly equal, to the critical value h_z=0. For these values of the parameters and for the number of spins considered one can appreciate the exponential dependence (linear in the semi-log scale). With a higher number of spins this dependence will change, saturating to the value R=1, as can be deduced from panel (b). Essentially, in this panel we focus on the rate of convergence of R from 1 to 0 in the neighborhood of h_z=0 as a function of number of spins N. Similarly, panel (g) shows the algebraic dependence of this rate as a function of N in the paramagnetic region. We have changed the caption of Fig 2 and the main text to make this point clearer.

Question 2: "- There seem to be a typo in Eq.(11): F should be F_Q."

Answer: We thank the referee for spotting this typo and we have corrected Eq. (11) in the current version of the paper.

Reply to Referee 2:

We would like to thank the referee for reviewing the manuscript and for his/her appreciation of our work. The referee is extremely positive on the originality and relevance of the work for the research community. He/she is "very inclined in recommending it for publication in SciPost".

In the following we will address point by point the remarks of the referee.

Remark 1: I understand that the authors evaluate the quantumness parameter R only for couple of parameters exploiting the property in Eq. (10). However I think it would be interesting to have the information also on the overall (three-parameter) quantumness. Is it always equal to the maximum of the two-parameter R that has been already evaluated?

Answer: We thank her/him for the interesting question. As can be deduced from the inequality (10), a 3-parameter quantumness is always lower-bounded by any 2-parameter quantumness obtained with a subset of the 3 parameters. In all the models examined we always observed a two-parameter quantumness which remains saturated at R=1 across the region of criticality (see R_zy Fig.2 (a) and R_xz in panel (e) for example). Since, R=1 is the maximum value of the quantumness, this immediately implies that the 3-parameter quantumness remains saturated to R=1 in all the models examined. Hence, on account of this trivial behaviour we have not included it in the work as it does not convey any relevant information. We added a paragraph to clarify this aspect in the current version of the manuscript.

Remark 2:I have found some of the figures (for example panels (d) and (h) of Fig. 2, or Fig. 4) not completely clear, or at least not self-explanatory. I think it would be more useful in the legend to make clear to what QFI matrix elements the points and the curves correspond to, instead of writing explicitly the results of the numerical fits (in particular I would suggest to put not much emphasis on the values of the intercepts values a of these linear fits).

Answer: We thank the referee her/him for the comments on the figures that helped us to present our work much more clearly. We will amend the legend to make the figures more self-explanatory, in accordance to the referee suggestion. We have edited the current version of the manuscript by adding a more clear reference to the QFI component.

Remark 3: Regarding to these fits, I have found some claims on the corresponding results not very clear/accurate. For example the authors write that in panel (d) one observes Heisenberg scaling, while in panel (h) a “normal scaling”. I think that these conclusions are drawn as the angular coefficients of the numerical linear regressions in the two panels are somehow close respectively to m=2 and m=1. However one does not obtain exactly these results and thus one should be more careful with these claims. For example also in panel (h) one obtains m=1.26 which is, I would say, definitely larger than m=1. In general I would suggest to the authors to — rephrase the corresponding sentences in the caption and in the main text — to add in these plots some lines with slopes corresponding to standard quantum limit scaling and Heisenberg scaling as a guide to the eye, in order to help the reader in observing the transition from one regime to the other one.

Answer: We are again grateful to the referee for his/her remark. In the coming version of the manuscript we have changed the text and the caption to a more accurate description of the scaling rate. We agree that the claim as it stated in the previous version may have lead to confusion. Following the suggestion of the referee we have added lines as a guide for the eye, with slopes corresponding to normal and Heisenberg scalings.

---

## Round 2 · List of Changes

- PG 4: We have corrected the typo in Eq. 11 according to the Referee 1 suggestion. We have also corrected another typo in Eq. 6.

- PG 9, line 22-26: The statement regarding the scaling behavior of the Quantum Fisher Information has been rephrased. The new sentence reads:
<< In panel (d) it is shown that in the ferromagnetic region, with hx near 1, both the x and y components of the QFI have a scaling very close to the Heisenberg limit, which allows to perform precise estimation in each direction. Otherwise, in the paramagnetic region (panel (h)) both the components of the QFI have a less enhanced scaling, closer to the normal quantum limit.>>

- PG 9, line 30-34: We have added a comment regarding the quantumness associated to the estimation of three parameters, which clarifies a point raised by the Referee 2: << It is worth mentioning that analyzing the quantumness associated with all the components of the magnetic field does not provide useful information for this system. Panel (a) and panel (e) of figure 2 show that the quantumness associated with at least one pair of components of magnetic fields is always maximal. In fact, from Eq. 10 it is straightforward to deduce that the quantumness of the complete set of parameters is always maximal, i.e. Rxyz=1.>>

- PG 10, Fig. 2 and PG 12, Fig. 4: In the figures we added lines, as guides for the eye, with slopes corresponding to Heisenberg and normal scalings. The caption has been rephrased to to reflect these changes and to provide a more detailed explanation of each panel.

- We have added Refs. 42,61,62.

- We have made some minor modifications to the acknowledgement and to the affiliations.

---

## Editorial Decision

published